# Measuring Variant-Specific Neutralizing Antibody Profiles after Bivalent SARS-CoV-2 Vaccinations Using a Multivariant Surrogate Virus Neutralization Microarray

**DOI:** 10.3390/vaccines12010094

**Published:** 2024-01-18

**Authors:** David Niklas Springer, Eva Höltl, Katja Prüger, Elisabeth Puchhammer-Stöckl, Judith Helene Aberle, Karin Stiasny, Lukas Weseslindtner

**Affiliations:** 1Center for Virology, Medical University of Vienna, A-1090 Vienna, Austriakatja.prueger@meduniwien.ac.at (K.P.); elisabeth.puchhammer@meduniwien.ac.at (E.P.-S.); judith.aberle@meduniwien.ac.at (J.H.A.); 2Center for Public Health, Medical University of Vienna, A-1090 Vienna, Austria; eva.a.hoeltl@meduniwien.ac.at

**Keywords:** SARS-CoV-2, Omicron, bivalent vaccination, neutralization, assay, surrogate

## Abstract

The capability of antibodies to neutralize different SARS-CoV-2 variants varies among individuals depending on the previous exposure to wild-type or Omicron-specific immunogens by mono- or bivalent vaccinations or infections. Such profiles of neutralizing antibodies (nAbs) usually have to be assessed via laborious live-virus neutralization tests (NTs). We therefore analyzed whether a novel multivariant surrogate-virus neutralization test (sVNT) (adapted from a commercial microarray) that quantifies the antibody-mediated inhibition between the receptor angiotensin-converting enzyme 2 (ACE2) and variant-specific receptor-binding domains (RBDs) can assess the neutralizing activity against the SARS-CoV-2 wild-type, and Delta Omicron BA.1, BA.2, and BA.5 subvariants after a booster with Omicron-adapted bivalent vaccines in a manner similar to live-virus NTs. Indeed, by using the live-virus NTs as a reference, we found a significant correlation between the variant-specific NT titers and levels of ACE2-RBD binding inhibition (*p* < 0.0001, r ≤ 0.78 respectively). Furthermore, the sVNTs identified higher inhibition values against BA.5 and BA.1 in individuals vaccinated with Omicron-adapted vaccines than in those with monovalent wild-type vaccines. Our data thus demonstrate the ability of sVNTs to detect variant-specific nAbs following a booster with bivalent vaccines.

## 1. Introduction

With the emergence of immune-evasive variants of the severe acute respiratory syndrome coronavirus type 2 (SARS-CoV-2), such as Delta and Omicron, the assessment of vaccine-induced humoral immunity has become increasingly complex. In addition, due to multiple previous infections with earlier variants and the sequential application of mono- and bivalent vaccines, the neutralizing activity of serum antibodies against those variants may vary considerably among individuals [1,2,3,4,5,6,7]. Consequently, the profiles of variant-specific antibodies cannot be adequately assessed by commercial assays that only measure antibody binding to the wild-type (WT) spike protein [8].

Live-virus neutralization tests (NTs) are considered the gold standard for analyzing neutralizing antibodies (nAbs) against multiple variants, including the Omicron subvariants (e.g., BA.1, BA.2, and BA.5) [1,2,3,5,6,9]. However, these tests are work intensive, slow, and require laboratories with high biosafety levels. Therefore, we and others evaluated variant-adapted surrogate-virus neutralization tests (sVNTs) that quantify the antibody-mediated binding inhibition of the viral receptor-binding domain (RBD) to its cellular receptor [angiotensin-converting enzyme 2 (ACE2)] [8,10,11]. 

In that regard, microarrays are particularly suitable as sVNT because the combination of multiple RBD proteins of different SARS-CoV-2 variants can be plotted as the target antigens into a single well, providing simultaneous measurement of antibody-mediated inhibition of ACE2 binding to the RBDs, which acts as a substitute for multiple variant-specific virus neutralization assays [8,10,11].

However, nAb titers against different SARS-CoV-2 variants span a wide concentration range, which are assessed by using dilution series of the serum samples in live-virus NTs. In contrast, in sVNTs, only a single dilution of the sample is used to measure inhibition. The concentration of antibodies against the RBD of one specific variant in one sample may already reach saturation and cause complete inhibition of RBD-ACE2 binding, whereas antibodies against the RBD of another variant may still be in an optimal quantification range.

Thus, multivariant sVNTs require thorough evaluation with multiple serial serum dilutions and correlation between the variant-specific measurements with the respective titers of live-virus NTs in order to identify correct cut-off concentrations within the sVNT’s linear test range.

To this end, we first evaluated a novel multivariant sVNT microarray using multiple variant-specific live-virus NTs as the reference. Then, we analyzed whether the increase in the breadth of variant-specific nAbs following Omicron-adapted vs. monovalent wild-type booster was similarly quantifiable by the sVNT as compared with live-virus NTs [6,12,13]. 

## 2. Materials and Methods

### 2.1. Sample Cohort

The study included 65 serum samples from our serum biobank of both SARS-CoV-2 vaccinated and infected individuals, as well as 30 additional pre-pandemic control samples from healthy individuals. The samples were drawn 15–40 days after the last vaccination or infection. 

The first cohort included samples from individuals hospitalized for a SARS-CoV-2 wild-type infection in the early pandemic before vaccines were available (n = 18, median age: 48 years (y), range: 22–77 y). The second and third cohorts included sera drawn approximately one month after the individuals’ respective second (n = 11, median age: 40 y, range 20–81) or third (n = 14, median age: 46 y, range: 27–64 y) mRNA monovalent wild-type vaccination. The fourth and fifth cohorts included individuals who received three monovalent wild-type vaccinations plus one bivalent Omicron-adapted BA.1/WT booster vaccination (n = 9, median age: 52 years, range: 38–62 y) or BA.5/WT (n = 13, median age: 50 y, range: 32–57 y). Only one sample was included from each individual. Detailed cohort characteristics are provided in Appendix A. 

All vaccinated individuals reported that they had not been infected with SARS-CoV-2, and anti-nucleocapsid antibodies were not detected in those cohorts (Anti-NC-IgG-ELISA, Euroimmun, Lübeck, Germany). None of the individuals reported any immunosuppressive therapy or condition. Data from the live-virus NTs were previously reported and were included only as reference for the sVNT evaluated in this study [2,6].

### 2.2. Live-Virus NTs

As previously described, each serum was tested for nAbs against the WT (with the D614G mutation), Delta, and Omicron BA.1, BA.2, and BA.5 SARS-CoV-2 subvariants [2,6].

In brief, SARS-CoV-2 variants were isolated from respiratory swabs taken from infected individuals using VeroE6 or VeroE6 TMPRSS2 cells (kindly provided by Anna Ohradanova-Repic). Virus sequences were determined using next-generation sequencing (Illumina, Minato, Tokyo) and uploaded to the GISAID database (WT, B.1.1 with the D614G mutation EPI_ISL_438123; Delta, B.1.617.2-like, sub-lineage AY.122:EPI_ISL_4172121; Omicron, B.1.1.529 + BA.*, sub-lineage BA.1.17:EPI_ISL_9110894; Omicron, B.1.1.529 + BA.*, sublineage BA.2:EPI_ISL_11110193; and Omicron, B.1.1.529 + BA.*, sub-lineage BA.5.3:EPI_ISL_15982848). The lineages were determined using Pango 4.1.3, Pango-data v.1.17.

Serum samples were serially diluted (two-fold) and incubated with 50–100 TCID50 SARS-CoV-2 for one hour at 37 °C. The dilution series ranged from 1:10 to 1:10,240. This mixture was then applied to VeroE6 cells and incubated for 3–5 days at 37 °C. Then, the cytopathic effect (CPE) was microscopically assessed, and the final titers were calculated as the inverse of the last titration at which CPE was prevented by serum-neutralizing activity. In all samples, the NTs were performed in duplicates.

### 2.3. Multivariant Surrogate-Virus Neutralization Test

The multivariant surrogate sVNT was performed using a similar protocol as in a previous study [8]. In brief, the basic framework of this assay is a commercial SARS-CoV-2 VoC ViraChip^®^ IgG microarray developed by Viramed (Planegg, Germany). However, in contrast to the standard version of the commercial microarray, the manufacturer plotted the RBD proteins of the SARS-CoV-2 wild-type (WT), the Delta variant, and the Omicron subvariants BA.1, BA.2, and BA.5 in triplets as the target antigens on the solid phase of each well, spatially separated into microspots.

After incubation of the wells with the diluted serum samples, recombinant ACE2 bound to alkaline phosphatase (ACE2-AP; also obtained from Viramed, Planegg, Germany) was added to each well, which could only bind to the RBD proteins in inverse correlation to the levels of neutralizing antibodies against the specific SARS-CoV-2 variants and Omicron subvariants present in the samples (until complete inhibition was achieved at the specific dilution). 

Finally, after a washing step, the bound ACE2-AP (if present) was made visible via a colorimetric reaction of a chromogen substrate and assessed using the Viramed plate reader. For each dilution step performed with each sample, the variant-specific ACE2-RBD binding inhibition was calculated as a percentage of the reduction in the inhibited color reaction relative to the maximum uninhibited color reaction obtained from a negative buffer control sample (incubation without serum but with ACE2-AP) after subtraction of the background (i.e., 100% represents complete inhibition of the binding).

Since neutralizing antibody titers against multiple variants in different serum samples covered a wide range (1:10 up to 1:2560), no single serum dilution was suitable to obtain valid ACE2-RBD inhibition values for all variants simultaneously. Therefore, each sample was tested in five serial two-fold dilutions, starting from 1:20 up to 1:320. In some instances, additional dilutions were required (up to 1:2560). Next, all values representing a total inhibition (=100%) were discarded. Additionally, all values below 20% were discarded for all dilutions except for the 1:20 one, since the preliminary data showed that this was below the assay’s linear range (and the error would have been amplified due to the dilutions). Then, we accounted for the additional dilution steps in the quantification and multiplied the corresponding values with the dilution factor relative to the starting dilution of 1:20. Finally, the result was the median of all valid dilution-corrected inhibition values for each variant. Appendix A provides a schematic overview of the described calculations.

### 2.4. Statistical Analysis

Statistical analyses were performed and figures were created with R version 4.2.0. Correlations between the NT titer and the sVNT results for all tested virus variants were calculated and reported as Spearman’s r (WT, Delta, BA.1, BA.2, and BA.5). Additionally, a Bland–Altman analysis was performed to assess the level of agreement between the two methods at different antibody concentrations [14].

Furthermore, cross-neutralizing activity was assessed by calculating the ratio of the WT neutralizing activity (log2 of the NT titer or sVNT result) to the variant-specific (Delta, BA.1, BA.2, and BA.5) NT titer or sVNT result (also log2 transformed) for each variant and each cohort. Therefore, NT ratios or sVNT ratios close to one represent strong cross-neutralization, whereas a value between zero and one represents the dominance of the WT activity over the variant-neutralizing activity. Finally, NT-ratios for each variant were compared between the cohorts using pairwise Wilcoxon ranked tests (multiplicity-adjusted for the cohorts using the Bonferroni–Holm method). Alpha was set to 0.05. Possible sVNT cut-off values that represent a positive NT titer (≥10) were calculated using ROC analysis and Youden’s index for each variant. 

## 3. Results

### 3.1. Correlation of the Multivariant sVNT with Variant-Specific Live-Virus NTs

In this study, we analyzed whether differences in the variant-specific neutralizing activity of antibodies induced by different vaccination regimes (mono- and bivalent SARS-CoV-2 mRNA vaccinations) could be assessed with a dilution-corrected multivariant sVNT similar to live-virus NTs.

Therefore, we first analyzed the correlation of variant-specific sVNT values to the respective NT titers assessed by live-virus NTs (Figure 1A). We observed a robust correlation for all SARS-CoV-2 variants, with a higher correlation for WT-, Delta-, and BA.2-specific neutralization than for Omicron BA.1 and BA.5 (Spearman: WT r = 0.87, Delta r = 0.90, BA.1 r = 0.79, BA.2 r = 0.87, and BA.5 r = 0.78). Then, we calculated cut-off values for the sVNT that correspond to positive NT titers (≥10) against the respective variants (Figure 1A, Appendix A).

Finally, we also conducted a Bland–Altman analysis [14] (Appendix A). The results of this analysis suggested a certain degree of non-linearity in the correlation between the two methods, as they indicated a trend that the sVNT results underestimated higher NT titer results and overestimated lower NT titers, although most of the sera were still within the bounds of the 95% agreement intervals.

### 3.2. Profiles of Neutralizing Activity after SARS-CoV-2 Wild-Type Infection and Vaccinations

Next, we analyzed neutralization profiles for the cohorts of WT-infected and vaccinated individuals (Figure 1B). The cohorts of WT-infected individuals and two-times vaccinated subjects showed moderate WT- and Delta-specific and weak Omicron-specific neutralizing activity. In contrast, higher levels of nAbs and a less pronounced difference between WT- and Omicron-specific neutralizing activities were observed for three-times vaccinated individuals and individuals who received a bivalent (WT/BA.1 or WT/BA.5) booster (fourth dose) vaccination. Indeed, median nAb levels against Omicron BA.2 and BA.5 were significantly higher in individuals who received a bivalent booster dose (WT/BA.1 or WT/BA5) as compared to subjects after WT-infection and individuals vaccinated only twice (*p* < 0.001 respectively). 

Notably, a BA.5-specific RBD-ACE2-binding inhibition, as assessed by the sVNT, was higher in all cohorts than for BA.2 and BA.1, which was inverse to the relation measured by the NT. 

### 3.3. Effect of Mono- and Bivalent Booster Vaccinations on the Neutralizing Activity

Finally, we analyzed the effect of monovalent versus bivalent booster vaccinations on the breadth of the neutralizing activity. To this end, we calculated the ratios of Delta, BA.1, BA.2, and BA.5 to the WT surrogate neutralization results, respectively. As shown in Figure 2, variant cross-neutralization (as expressed by ratios closer to one) was increased after the third monovalent vaccination compared to only two-times vaccinated or WT-infected individuals. Indeed, the three-times vaccinated individuals had significantly higher Omicron BA.2- and BA.5-neutralizing activities than those after two vaccinations, with median ratios closer to one (for *p*-values, see Appendix A).

A further increase in the neutralization breadth was observed in individuals with a bivalent vaccination compared to those with monovalent vaccinations (Figure 2). Notably, the bivalent BA.5/WT-vaccinated individuals had significantly higher Omicron BA.1 and BA.5 cross-neutralizing activity than all other cohorts except for the bivalent WT/BA.1-vaccinated individuals who exhibited higher Delta and BA.2 cross-neutralization than the two-times vaccinated and wild-type-infected cohorts. Similarly, the BA.1/WT bivalent-vaccinated cohort displayed more Delta and BA.2 cross-neutralization than the WT infected and two-times vaccinated individuals, as well as more BA.1 cross-neutralization than the two- or three-times monovalent vaccinated cohorts.

## 4. Discussion

In this study, we evaluated whether a novel multivariant sVNT that is based on a commercial microarray and quantifies the antibody-mediated inhibition of binding between ACE2 and variant-specific RBD proteins can identify differences in the cross-neutralization after monovalent versus bivalent booster vaccinations similarly to variant-specific live-virus NTs. 

Indeed, in earlier studies that used live-virus NTs, individuals vaccinated three times with monovalent mRNA WT vaccines showed a significantly broader neutralizing activity than WT-infected or twice-vaccinated subjects [15,16]. However, no further improvement in the cross-neutralizing activity was found after a fourth monovalent wild-type vaccine dose in comparable cohorts [6,17,18,19,20]. Hence, bivalent vaccines containing mRNA encoding both the SARS-CoV-2 WT, as well as Omicron BA.1 or BA.5, have been developed to increase the (cross)-neutralizing antibody activity against circulating and possibly future variants [21]. 

Thus, measuring cross-neutralizing antibody profiles after bivalent booster vaccinations may be essential, e.g., for vaccine efficacy studies and routine diagnostics. However, this is limited by the laborious setup of performing multiple different variant-specific live-virus NTs with the same samples (e.g., [2,3,4,5,6]). Using this labor-intensive approach, our group previously reported that bivalent vaccination moderately increased neutralizing activity against the respective Omicron variant applied with the vaccine [6].

In the present study, we used the multivariant sVNT and similarly demonstrated that the bivalent BA.1/WT-boosted cohort displayed increased cross-neutralization of BA.1 compared to the individuals three-times vaccinated with monovalent vaccines. Also, similar to the live-virus NTs, individuals vaccinated with BA.5/WT-adapted vaccines showed an increased cross-neutralization of both BA.1 and BA.5. Yet, in both bivalent vaccinated cohorts, wild-type titers were generally higher than Omicron titers, which has been attributed to immune imprinting to WT [22].

Wild-type-infected and twice-vaccinated individuals showed similar cross-neutralization (Figure 2), as also seen in earlier studies [2,3,23]. Unexpectedly, the sVNT detected stronger cross-neutralization against BA.1 in the WT-infected cohort than in individuals vaccinated twice, but this may be related to the large number of individuals with low nAb levels against BA.1 in the latter cohort. Test-inherent differences, in the sensitivity of detecting nAbs against different Omicron subvariants by the sVNT, might also contribute to this difference that we observed. Furthermore, the sVNT generally measured a higher neutralizing activity against BA.5 than against BA.2, which was inverse to NT titer results, as reported earlier [6]. However, the relation of neutralizing activity against different variants may also vary between different neutralization assays (e.g., depending on the kinetics of virus growth), as greater BA.5 than BA.2 NT titers in vaccinated individuals have also been reported previously [5]. 

If antibody neutralization against different Omicron subvariants is not assessed with a similar sensitivity by the sVNT microarray (e.g., due to differences in the concentration or conformation of the RBDs), this might pose a limitation, complicating a comparison of quantitative test results by the sVNT with the respective NT titers. While the issue of assay oversaturation at high antibody concentrations can be overcome by performing dilution series, such possible test-inherent differences in sensitivity would require fine tuning the microarray platform, such as applying different antigen concentrations. However, in this study, we evaluated a commercially available immunoassay that served as a basis for the sVNT, wherefore such adaptations were beyond the scope of this study. Another limitation of this study, particularly with regard to comparing different vaccination regimens, is the modest sample size for the different cohorts. However, this study mainly served as a proof-of-concept of which results similar to those obtained by live-virus NTs [5,6,12,13] are also reproducible with a lower-effort sVNT. 

Overall, the sVNTs gave very similar results to the live-virus NTs, which enables an analysis of cross-neutralizing antibodies against different SARS-CoV-2 variants in seroprevalence studies involving large parts of the population. Indeed, due to individual vaccination and infection histories, the profiles of nAbs against different variants and Omicron subvariants in the population will be highly heterogeneous, as has been demonstrated by Zaballa and colleagues [10].

For this reason, future studies investigating the benefits of adapting COVID-19 vaccines to newly circulating strains (such as XBB) might measure antibody profiles in large study cohorts using sVNTs that allow for a higher and faster turnover than live-virus NTs. 

Thus, our data evaluating a sVNT microarray, including multiple dilution steps to identify optimal cut-offs with variant-specific live-virus NTs as the reference, may be relevant for applying such immunoassays in vaccine efficacy studies. 

## 5. Conclusions

In this study, we demonstrate that not only laborious variant-specific live-virus NTs but also a more feasible multi-variant sVNT can identify distinct profiles of nAbs against different SARS-CoV-2 variants and Omicron sub-variants. Thus, the multivariant sVNT was able to assess the effect of Omicron-adapted SARS-CoV-2 mRNA vaccines on the breadth of nAbs against different variants.

## Figures and Tables

**Figure 1 vaccines-12-00094-f001:**
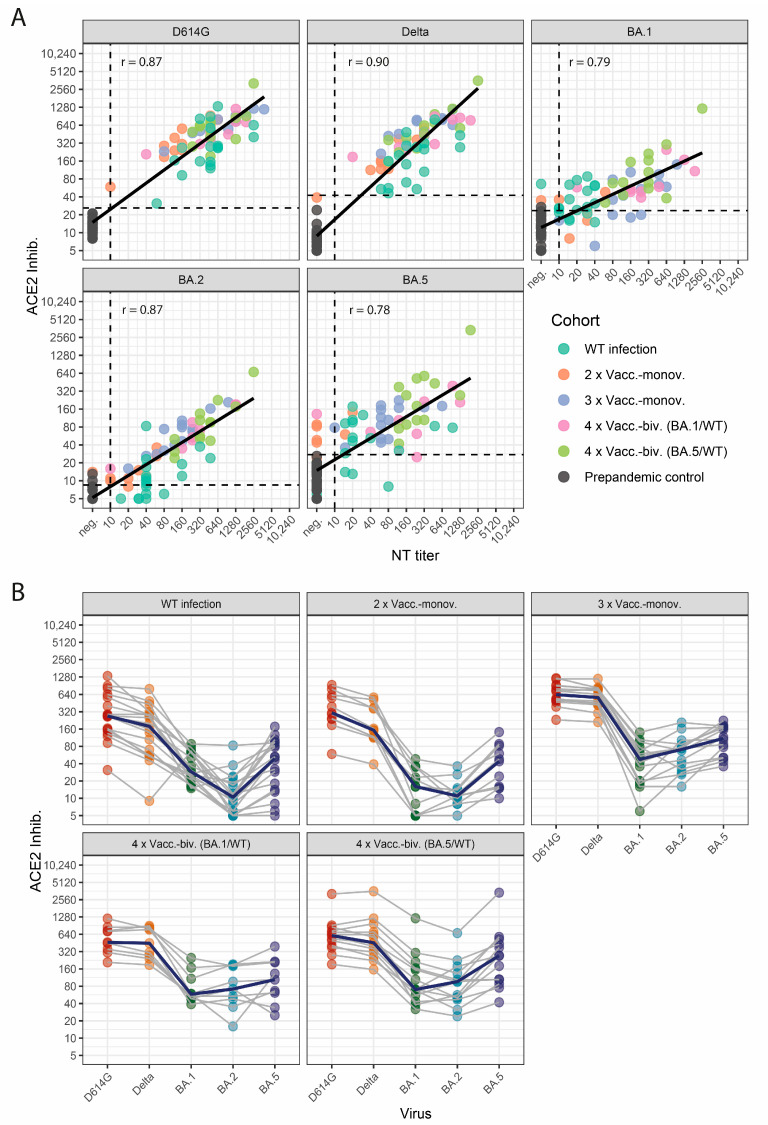
Correlation of sVNT values to NT titers against D614G (WT), Delta, and Omicron BA.1, BA.2, and BA.5. (**A**) Each dot represents a single serum (n = 95). The *x*-axis shows live-virus NT titers; the *y*-axis shows the neutralizing activity assessed by the sVNT ACE2-Inhib. (ACE2-RBD inhibition). The cohorts are color coded as indicated. Spearman’s r is indicated in the top-left corner of each panel. The dotted lines indicate the cutoff values of the respective tests. The black line indicates the linear regression slope. (**B**) Neutralization profiles of vaccinated and infected cohorts. Each serum is represented by five dots (one for each variant), connected by a grey line. The bold blue lines indicate the median titer for each variant. WT infection: sera drawn from unvaccinated patients hospitalized for a wild-type SARS-CoV-2 infection (n = 18). 2 × Vacc.-monov./3 × Vacc.-monov. Sera drawn approximately one month after two (n = 11)/three vaccinations (n = 14) with a monovalent wild-type vaccine. 4 × Vacc.-biv. (BA.1/WT)/4 × Vacc.-biv. (BA.5/WT): Sera were drawn after a booster with a bivalent Omicron-adapted vaccine containing mRNA encoding for the Spike of either BA.1 and WT (n = 9) or BA.5 and WT (n = 13).

**Figure 2 vaccines-12-00094-f002:**
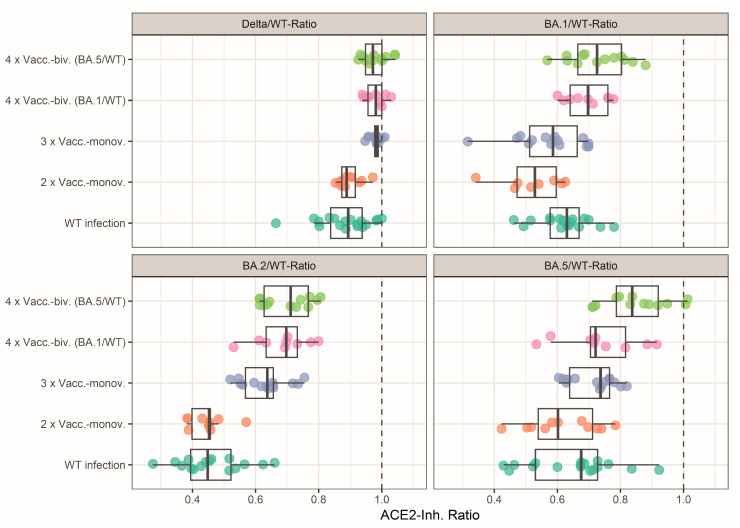
Neutralizing activity of the vaccinated and infected cohorts. Cross-neutralizing activity was assessed by calculating the ratio of sVNT values (ACE2-Inhib.) of Delta, BA.1, BA.2, and BA.5 divided by the ACE2-Inhibition against WT, respectively. A single dot represents one serum. The median is shown as a solid line, and boxes and whiskers represent the quartiles and range, respectively. ACE2-Inhib. (ACE2-RBD Inhibition). WT infection: sera drawn from unvaccinated patients hospitalized for a wild-type infection (n = 18). 2 × Vacc.-monov./3 × Vacc.-monov. Sera drawn approximately one month after two (n = 11)/three vaccinations (n = 14) with a monovalent wild-type vaccine. 4 × Vacc.-biv. (BA.1/WT)/4 × Vacc.-biv. (BA.5/WT): Sera were drawn after a booster with a bivalent Omicron-adapted vaccine containing mRNA encoding for the Spike of either BA.1 and WT (n = 9) or BA.5 and WT (n = 13). The cohorts are color coded as in Figure 1A.

## Data Availability

Data from this study has not been publicly archived.

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
