# Peer review of "Measuring Variant-Specific Neutralizing Antibody Profiles after Bivalent SARS-CoV-2 Vaccinations Using a Multivariant Surrogate Virus Neutralization Microarray"

_vaccines, 2024, doi:10.3390/vaccines12010094_

Round 1
Reviewer 1 Report
Comments and Suggestions for Authors
The study by Springer et al. investigates whether a multivariate surrogate neutralization test (vNT) for SARS CoV-2 produces similar results to neutralization tests (NT) with infectious viruses. With vNT, values for the neutralization of several viruses are obtained simultaneously in one measurement.
The study initially shows that the serum neutralization titers of vNT and NT correlate well with individual SARS CoV-2 virus variants. In addition, the study examined sera from five groups of test subjects that differ in terms of infection and the number and type of vaccinations against SARS CoV-2. When comparing the measured values in the different groups, the study comes to the conclusion that the test can differentiate the immune response in the different groups in a similar way to the classic NT. The authors conclude that the multivariate vNT could be used instead of the NT in vaccination studies in the future.
The study makes an important contribution to the simplification of SARS CoV-2 neutralization tests.
Questions and comments:
Methods and material:
Line 98: Serum samples...: Was the test performed in single or multiple measurements?
Lines 122-124: Why was buffer used as a negative control and not an antibody negative serum? Is there data on background titer values for antibody-negative sera?
Results:
Line 172: Supplementary Table S3 is probably meant instead of S1.
Line 190, In contrast, higher levels ...: was the difference between the groups statistically significant?
Figure 1: the colors used in the figure are sometimes difficult to distinguish.
Line 198: this probably refers to Supplementary Table S4 instead of S3.
Author Response
Response to the Reviewers’ comments:
We are very grateful for the rapid revision of our manuscript and the valuable Reviewers’ comments, which improved the manuscript significantly. Following the Reviewers’ recommendations, we conducted a thorough, major revision of the paper, as recommended. Changes in the text are highlighted in yellow (marked version). Indicated numbers of pages, lines and figures in this response refer to the marked-up version of the submitted manuscript.
Reviewers’Comments to the Author:
Reviewer #1
Comments to the Author
The study by Springer et al. investigates whether a multivariate surrogate neutralization test (vNT) for SARS CoV-2 produces similar results to neutralization tests (NT) with infectious viruses. With vNT, values for the neutralization of several viruses are obtained simultaneously in one measurement.
The study initially shows that the serum neutralization titers of vNT and NT correlate well with individual SARS CoV-2 virus variants. In addition, the study examined sera from five groups of test subjects that differ in terms of infection and the number and type of vaccinations against SARS CoV-2. When comparing the measured values in the different groups, the study comes to the conclusion that the test can differentiate the immune response in the different groups in a similar way to the classic NT. The authors conclude that the multivariate vNT could be used instead of the NT in vaccination studies in the future.
The study makes an important contribution to the simplification of SARS CoV-2 neutralization tests.
Response:
We thank the reviewer for the encouraging comment.
Questions and comments:
- Methods and material: Line 98: Serum samples...: Was the test performed in single or multiple measurements?
Response:
Indeed, the test was performed in duplicates in all sampes. This information is now included in the manuscript (changes in line 102).
- Lines 122-124: Why was buffer used as a negative control and not an antibody negative serum? Is there data on background titer values for antibody-negative sera?
Results:
For this study, we applied the sVNT using the protocol the manufacturer the recommended. This protocol suggested the use of buffer rather than control sera, since at it is easier to standardize than Anti-SARS-CoV-2-antibody-negative sera. However, we also included a pre-pandemic control cohort to evaluate the test´s specificity in performance in Anti-SARS-CoV-2-antibody negative sera. The results of these measurements are shown in Figure 1A, color coded in grey. As shown in this figure, we also calculated a possible cutoff-values based on these negative sera (dashed line in Figure 1A), which e.g. indicated a cut off value of 26 arbitrary units in the sVNT for neutralizing titers of at least 10 in the wildtype NT.
- Line 172: Supplementary Table S3 is probably meant instead of S1.
Response:
We thank the Reviewer for finding ths typo, which we now corrected in the revised version of the manuscript (changes in line 164).
- Line 190, In contrast, higher levels ...: was the difference between the groups statistically significant?
Response:
Indeed, the median nAb levels against Omicron BA.2 and BA.5 were significantly higher in individuals who received a bivalent booster dose (WT/BA.1 or WT/BA5) compared to subjects after WT-infection and individuals vaccinated only twice (p<0.001 respectively). This information is now included in the revised version of the manuscript (changes in lines 191-194).
- Figure 1: the colors used in the figure are sometimes difficult to distinguish.
Response:
We thank the Reviewer for this comment! We now changed the color coding to a palette that allows for a better visual discrimination (changes in Figure 1A and Figure 2).
- Line 198: this probably refers to Supplementary Table S4 instead of S3.
Response:
Again, we thank the Reviewer for finding this typo, which we now corrected (changes in lines 206).
Reviewer 2 Report
Comments and Suggestions for Authors
Reject
Author Response
Response to the Reviewers’ comments:
We are very grateful for the rapid revision of our manuscript and the valuable Reviewers’ comments, which improved the manuscript significantly. Following the Reviewers’ recommendations, we conducted a thorough, major revision of the paper, as recommended. Changes in the text are highlighted in yellow (marked version). Indicated numbers of pages, lines and figures in this response refer to the marked-up version of the submitted manuscript.
Reviewers’Comments to the Author:
Reviewer #2
No specific Comments to the Authors
Response:
We acknowledge that the Reviewer did not indicate major shortcomings in the introduction, the references, the research design, the methods, and the conclusions but finally recommended “Rejection”. We hope that the thorough revision of our manuscript improved the paper to a point that the Reviewer might change this recommendation.
Reviewer 3 Report
Comments and Suggestions for Authors
The manuscript by Springer et al describes an improved surrogate neutralisation assay (sVNT) to detect the antibody response to SARS-CoV-2, especially for the variants of concern. The manuscript is well written and easy to follow. I have no issue with some of the conclusions:
the sVNT can detect the nAb against VOC and 4 vaccinations with the last one being a bivalent vaccine generates a higher anti-Omicron responses than a 3x monovalent vaccination regime.
My main issue is with the methodology behind the analysis of the data, specifically the "adjustment factor". Specifically,
1) how is meant to be used? If a different set of samples are tested the correlation between NT and sVNT could be different, and therefore the adj factor will be different. Specifically, most clinical samples will likely be from individuals vaccinated and infected, but this set of samples are not included in the study.
2) why the adjustment factor is needed in first place? If I understood correctly the aim is to have the results from sVNT more similar to the NT results. However, it may mislead the results. For example in line 194 authors stated that responses to BA.5 were higher than BA.1 and BA.2 and likely to be test-inherent and required correction. However, I could find a paper in which responses to BA.2 are lower than BA.5 for convalescent individuals and monovalent vaccinated: fig 1 in https://doi.org/10.1038/s41467-023-41049-4. The NT with different variants will have differences based on kinetic of the virus growth, and it is difficult to compare results between variants. A better approach would be to have a reference preparation such as an International Standard and report the results in International Units, relative to the standard. This could be beyond the scope of the work if the aim is to prove that the sVNT is able to detect antibody direct against VOC antigen, but I would drop the adjustment factor instead.
Two minor points:
1) in the discussion it should mentioned that the limit of the study is that there is no group with 4 monovalent vaccinations. The authors add a reference to support the concept that 4 monovalent vaccination do not broad the antibodies reactivity to VOC, but it would have been nice to have shown this with the methods in this manuscript
2) a better statistical comparison between methods is achieved using a Bland-Altman analysis or a Deming regression rather than a Spearman correlation
Comments on the Quality of English Language
Overall, no issue with the English, the manuscript reads well. There are some typos here and there, like the (ref?) in the discussion line 256 which needs to be addressed.
Author Response
Response to the Reviewers’ comments:
We are very grateful for the rapid revision of our manuscript and the valuable Reviewers’ comments, which improved the manuscript significantly. Following the Reviewers’ recommendations, we conducted a thorough, major revision of the paper, as recommended. Changes in the text are highlighted in yellow (marked version). Indicated numbers of pages, lines and figures in this response refer to the marked-up version of the submitted manuscript.
Reviewers’Comments to the Author:
Reviewer #3
The manuscript by Springer et al describes an improved surrogate neutralisation assay (sVNT) to detect the antibody response to SARS-CoV-2, especially for the variants of concern. The manuscript is well written and easy to follow. I have no issue with some of the conclusions: the sVNT can detect the nAb against VOC and 4 vaccinations with the last one being a bivalent vaccine generates a higher anti-Omicron responses than a 3x monovalent vaccination regime.
My main issue is with the methodology behind the analysis of the data, specifically the "adjustment factor". Specifically,
- how is meant to be used? If a different set of samples are tested the correlation between NT and sVNT could be different, and therefore the adj factor will be different. Specifically, most clinical samples will likely be from individuals vaccinated and infected, but this set of samples are not included in the study.
- why the adjustment factor is needed in first place? If I understood correctly the aim is to have the results from sVNT more similar to the NT results. However, it may mislead the results. For example in line 194 authors stated that responses to BA.5 were higher than BA.1 and BA.2 and likely to be test-inherent and required correction. However, I could find a paper in which responses to BA.2 are lower than BA.5 for convalescent individuals and monovalent vaccinated: fig 1 in https://doi.org/10.1038/s41467-023-41049-4. The NT with different variants will have differences based on kinetic of the virus growth, and it is difficult to compare results between variants. A better approach would be to have a reference preparation such as an International Standard and report the results in International Units, relative to the standard. This could be beyond the scope of the work if the aim is to prove that the sVNT is able to detect antibody direct against VOC antigen, but I would drop the adjustment factor instead.
Response:
We thank the Reviewer for this valuable comment. Indeed, the rationale of calculating adjustment factors was to compensate for a possible systematic variation, that was indicated by the differences we observed for the sVNT´s sensitivity for different variants. However, we completely agree with the Reviewer that we cannot exclude that these observed differences might be associated with the included cohorts. Furthermore, the Reviewer is correct that additional validation would be required, which our study could not provide.
As recommended by the Reviewer, we therefore completely removed the calculations for the adjustment factor from the manuscript. Furthermore, we included a section in the manuscript that discusses that not only the sVNT but also the NT might differ in the sensitivity to detect neutralizing antibodies gainst different variants. We thank the reviewer to pointing to this reference, which as also included in the discussion.
Changes in lines: 196-197, 259-266 (and deletions of the paragraphs and Supplementary Figures/Tables concerning the adjustment factor).
Two minor points:
- in the discussion it should mentioned that the limit of the study is that there is no group with 4 monovalent vaccinations. The authors add a reference to support the concept that 4 monovalent vaccination do not broad the antibodies reactivity to VOC, but it would have been nice to have shown this with the methods in this manuscript
Response:
While we agree with the Reviewer that inclusion of a cohort with monovalent vaccinations would have been a good approach, unfortunately we did not have a suitable cohort available (or rather, not enough sample material left of four times vaccinated individuals included in previous studies). However, we included additional references in the revised version manuscript that underlined the concept the Reviewer indicates.
- a better statistical comparison between methods is achieved using a Bland-Altman analysis or a Deming regression rather than a Spearman correlation
Response:
As suggested by the Reviewer, we now included a Bland-Altman analysis in the manuscript and present the results in Supplementary Figure 2. The method and the results are furthermore included in the manuscript (changes in lines 139-141, 165-169).
Reviewer 4 Report
Comments and Suggestions for Authors
The authors examined the performance of a surrogate virus neutralization test in a multivariant microarray format and compared it with the standard live virus neutralization test, using samples from cohorts with distinct immunological history of SARS-CoV-2 exposure. They provide proof that such an approach can provide an alternative to neutralization tests and could be used in vaccine efficacy studies.
They use low sample numbers (9 to 18) per cohort but yet observe cohort-specific patterns. They attribute some of their observations to the measurement system (test-inherent) and try to eliminate such effects by mathematical corrections.
The idea that the inhibitory effect of serum antibodies upon ACE2-RBD interactions would reflect variant-specific subtleties is straightforward, such a system should work if properly adjusted. The authors focus on adjusting serum dilutions and calculations in order to optimize their assay, while the properties of the microarray are not adjusted. Accordingly, the authors note that concentrations and conformations of the RBD on the microarray might be responsible for some of their observations.
Perhaps these weaknesses should be discussed and solutions proposed in the last section of the article.
minor remarks
Color coding in figure 1A is very hard to differentiate.
Line 256 (refs?) - remove of insert the intended references
Author Response
Response to the Reviewers’ comments:
We are very grateful for the rapid revision of our manuscript and the valuable Reviewers’ comments, which improved the manuscript significantly. Following the Reviewers’ recommendations, we conducted a thorough, major revision of the paper, as recommended. Changes in the text are highlighted in yellow (marked version). Indicated numbers of pages, lines and figures in this response refer to the marked-up version of the submitted manuscript.
Reviewers’Comments to the Author:
Reviewer #4
The authors examined the performance of a surrogate virus neutralization test in a multivariant microarray format and compared it with the standard live virus neutralization test, using samples from cohorts with distinct immunological history of SARS-CoV-2 exposure. They provide proof that such an approach can provide an alternative to neutralization tests and could be used in vaccine efficacy studies.
They use low sample numbers (9 to 18) per cohort but yet observe cohort-specific patterns. They attribute some of their observations to the measurement system (test-inherent) and try to eliminate such effects by mathematical corrections.
The idea that the inhibitory effect of serum antibodies upon ACE2-RBD interactions would reflect variant-specific subtleties is straightforward, such a system should work if properly adjusted. The authors focus on adjusting serum dilutions and calculations in order to optimize their assay, while the properties of the microarray are not adjusted. Accordingly, the authors note that concentrations and conformations of the RBD on the microarray might be responsible for some of their observations.
Perhaps these weaknesses should be discussed and solutions proposed in the last section of the article.
Response:
We thank the Reviewer for this valuable comment! We now included a text segment on the discussion of the additional limitations, including the the modest sample sizes, and that the properties of the microarray were not adjusted (as the microarray was obtained from a commercial manufacturer). We also removed some of the calculations, as has been suggested by Reviewer 2. Changes in the text in lines 270-279.
minor remarks
- Color coding in figure 1A is very hard to differentiate.
Response:
Thnak you very much for this comment! Following the Reviewer´s recommendation, we now changed the color scheme to a palette that allows for a better visual discrimination.
- Line 256 (refs?) - remove of insert the intended references
Response:
We thank the reviewer for pointing out this oversight. The missing references have now been included in the revised version of the manuscript (line 256).
Round 2
Reviewer 3 Report
Comments and Suggestions for Authors
The authors have addressed in a satisfactory manner my concerns. I don't have any further comments.